# Galectin-4 Is Involved in the Structural Changes of Glycosphingolipid Glycans in Poorly Differentiated Gastric Cancer Cells with High Metastatic Potential

**DOI:** 10.3390/ijms241512305

**Published:** 2023-08-01

**Authors:** Kazuko Hachisu, Akiko Tsuchida, Yoshio Takada, Mamoru Mizuno, Hiroko Ideo

**Affiliations:** 1Laboratory of Glyco-Organic Chemistry, The Noguchi Institute, 1-9-7, Kaga, Itabashi, Tokyo 173-0003, Japan; khirose@noguchi.or.jp (K.H.); mmizuno@noguchi.or.jp (M.M.); 2Laboratory of Glycobiology, The Noguchi Institute, 1-9-7, Kaga, Itabashi, Tokyo 173-0003, Japan; akikots@noguchi.or.jp (A.T.); takada@noguchi.or.jp (Y.T.)

**Keywords:** galectin-4, gastric cancer, peritoneal dissemination, glycosphingolipid, glycan, glycomics, lectin

## Abstract

Gastric cancer with peritoneal dissemination is difficult to treat surgically, and frequently recurs and metastasizes. Currently, there is no effective treatment for this disease, and there is an urgent need to elucidate the molecular mechanisms underlying peritoneal dissemination and metastasis. Our previous study demonstrated that galectin-4 participates in the peritoneal dissemination of poorly differentiated gastric cancer cells. In this study, the glycan profiles of cell surface proteins and glycosphingolipids (GSLs) of the original (wild), galectin-4 knockout (KO), and rescue cells were investigated to understand the precise mechanisms involved in the galectin-4-mediated regulation of associated molecules, especially with respect to glycosylation. Glycan analysis of the NUGC4 wild type and galectin-4 KO clones with and without peritoneal metastasis revealed a marked structural change in the glycans of neutral GSLs, but not in *N*-glycan. Furthermore, mass spectrometry (MS) combined with glycosidase digestion revealed that this structural change was due to the presence of the lacto-type (β1-3Galactosyl) glycan of GSL, in addition to the neolacto-type (β1-4Galactosyl) glycan of GSL. Our results demonstrate that galectin-4 is an important regulator of glycosylation in cancer cells and galectin-4 expression affects the glycan profile of GSLs in malignant cancer cells with a high potential for peritoneal dissemination.

## 1. Introduction

Gastric cancer (GC) is one of the most common cancers and is the fourth leading cause of cancer-related deaths worldwide and incidence rates are highest in eastern Asia [1]. Peritoneal dissemination is the most common form of metastatic or recurrent GC and a major cause of increased mortality. It is difficult to treat surgically, and sufficiently effective treatment has not yet been established [2].

Monoclonal antibodies and immune checkpoint inhibitors that target molecules such as HER2, VEGFR, and PD-1 are being used for GC, but are not effective in all patients [3]. Therefore, it is vital to clarify the mechanism of peritoneal metastasis for the early diagnosis and treatment of GC with recurrence and metastasis. Peritoneal dissemination is a process in which cancer cells detach from neighboring cells and adhere to the peritoneum; therefore, it is necessary to understand the molecules involved in cell–cell interactions, invasion, and proliferation [4].

Galectins, a family of lectins that recognize glycans with β-galactoside, have been reported to change their expression in response to cancer development in various tissues [5]. Galectin-4 is expressed in the epithelial cells of the normal gastrointestinal tract and has two glycan-binding domains with distinct binding specificities. Accordingly, galectin-4 may function by crosslinking molecules and regulating several biological processes [6,7]. Galectin-4 has been detected in many cancer types; however, it plays contradictory roles in different types of cancer cells [8,9,10,11,12,13]. This characteristic of galectin-4 may be due, in part, to differences in the factors surrounding cancer tissues and cells, especially the molecules and glycans that interact with galectin-4 [14].

In our previous study, we observed that galectin-4, but not galectin-3, was highly expressed in poorly differentiated gastric cancer cells with high metastatic potential. Using knockout and knockdown techniques, we revealed that galectin-4 participates in the peritoneal dissemination of malignant gastric cancer cells by promoting cell proliferation and interacting with several molecules, including c-MET and CD44 [15].

In the present study, we performed a detailed analysis of glycans to elucidate the precise mechanisms involved in the galectin-4-mediated regulation of associated molecules. This is the first study that investigates the glycan profiles of cell surface proteins and glycosphingolipids (GSLs) of the cells with different galectin-4 expression and metastatic potential simultaneously.

## 2. Results

### 2.1. N-Glycan Structure of Membrane Proteins

To elucidate the mechanisms involved in galectin-4-mediated regulation with respect to glycosylation, we first performed *N*-glycomics of membrane proteins (hydrophobic fraction) of wild-type and galectin-4 KO clones of the poorly differentiated gastric cancer cell line NUGC4. *N*-glycans were released by peptide *N*-glycosidase F (PNGase F) and were *O*-benzyloxyamine (BOA)-labeled using the glycoblotting method. The matrix-assisted laser desorption/ionization time-of-flight mass spectrometry (MALDI-TOF MS) results (*m*/*z* 2020–4000) showed that highly fucosylated, bisecting GlcNAc, and highly branched complex-type glycans with galactose (Gal), such as G2F2+GN, G2F3+GN, G3F3+GN, G4F2+GN, G4F3+GN, and G4F4+GN, were detected in wild-type and KO NUGC4 cells (Figure 1) (abbreviations of glycans are listed in Appendix A). The detailed structures of some *N*-glycans were identified using MALDI-mass spectrometry/mass spectrometry (MS^2^) (Appendix A). However, no significant differences in *N*-glycan profiles were observed between wild-type and KO clones.

### 2.2. Glycan Structure of Sulfated GSLs

As there were no significant differences in the *N*-glycan profiles between the wild type and the KO, we focused our attention on GSLs. We first analyzed acidic GSLs, including sulfated GSLs, using high-performance thin-layer chromatography (HPTLC) and MALDI-TOF MS. SM4 was mainly observed in both analyses. MS analysis also showed trace amounts of SM3 and SM2 peaks. Sialylated GSLs were undetectable in MS spectra (*m*/*z* 750 to 1400) (Figure 2). The ceramide moiety was mainly identified by the d18:1-C16:0 (SM4 *m*/*z* 778) and d18:1-C24:1h (SM4 *m*/*z* 904) backbones (Figure 2). However, no significant difference in acidic GSLs was observed between wild-type and KO NUGC4 cells (Figure 2 and Appendix A).

### 2.3. Glycan Structure of GSLs

We observed differences in neutral GSLs between the wild type and the KO in preliminary experiments. Therefore, we also analyzed GSLs from control cells (Con) and galectin-4 re-expressing cells (Res). The extracted GSLs were subjected to HPTLC, and several bands of neutral GSLs significantly increased in the KO compared to the wild type, Con, and Res (Figure 3a). To identify the increase in neutral GSLs, MALDI-TOF MS analysis of the neutral GSL glycans was performed by releasing the glycans using endoglycosylceramidase (EGCase). In Figure 3b, the peak signals at *m*/*z* 835, 981, 1038, and 1184 were assigned to H3N1, H3N1F1, H3N2, and H3N2F1, respectively (H, hexose; N, *N*-acetylhexosamine; F, fucose). H3N1 and H3N1F1 were detected as the main peaks, and other peaks were negligible in the wild-type, Con, and Res cells, whereas H3N1, H3N1F1, H3N2, and H3N2F1 were detected in almost equal proportions in the KO showing a significant increase in glycans H3N1, H3N2, and H3N2F (Figure 3b). We also analyzed GSL glycans without EGCase treatment and found no free oligosaccharides in the NUGC4 cells.

MALDI-TOF MS analysis of the total GSL glycans revealed that the structures were consistent with those of the aforementioned neutral GSLs. 

GSLs are classified into lacto (Lc), neolacto (nLc), ganglio (Gg), globo (Gb), and isoglobo (iGb) types, each containing a different set of enzymes involved in their biosynthetic pathway. Because sialylated glycans are undetectable in GSLs of NUGC4 cells, they are restricted to the Lc/nLc or Gb/iGb types (Appendix A). 

### 2.4. Detailed Structural Identification of GSL Glycans in NUGC4 Cells (Wild and Galectin-4KO) Using Glycosidase and MS^n^ Analysis

To identify the detailed glycan structures of GSLs that were upregulated in the KO, we performed MS and MS^n^ analyses combined with glycosidase digestion using 2-aminobenzamide (2AB) labeling in the glycoblotting method (Figure 4 and Appendix A). As illustrated in Figure 4a, labeled GSL glycans were cleaved with α1-3/4Fucosidase in Step 1, and fucosidase-treated glycans were digested by β1-4Galactosidase and/or β1-3Galactosidase in Step 2.

To elucidate the detailed structures of H3N1F1 (*m*/*z* 996), we first acquired the reference MALDI-MS^2^ spectra of five Lc/nLc-type standards (STDs) with 2AB labeling, which are candidates for this glycan (Appendix A). Subsequently, MS^2^ analysis of the H3N1F1 glycan from NUGC4 cells was performed, and the structure of H3N1F1 was determined to be lacto-*N*-fucopentaose (LNFP) II (β1-3Gal, Le^a^) or III (β1-4Gal, Le^x^) (Appendix A). Because LNFP II and III STDs showed similar fragment patterns in the MS^3^ analysis (Appendix A), we digested them with an enzyme (β1-3 or β1-4Galactosidase) for identification. 

Since adjacent α-fucose molecules prevent β-galactosidase digestion, we first cleaved the fucose of H3N1F with α1-3/4Fucosidase in Step 1 (Figure 4b). Then, in Step 2-1, fucosidase-treated glycans (H3N1, *m*/*z* 850) were digested by β1-4Galactosidase, which showed almost complete cleavage in the wild type, indicating that the structure of H3N1F1 (*m*/*z* 996) in the wild type was LNFP III (β1-4Gal, Le^x^). In contrast, 40% of the glycan (H3N1, *m*/*z* 850) in the KO was not digested by β1-4Galactosidase (Figure 4c). This undigested glycan was confirmed to be the β1-3Galactosylated glycan because it was digested by β1-3Galactosidase in Step 2-2 (Figure 4d). Thus, approximately 60% of H3N1F1 in the KO was found to be nLc-type LNFP III (β1-4Gal, Le^x^), and 40% residual H3N1F1 was found to be Lc-type LNFP II (β1-3Gal, Le^a^). These results were validated using glycan STDs (LNFP II and III) as controls to confirm digestion. 

The detailed structures of H3N2 (*m*/*z* 1053) and H3N2F1 (*m*/*z* 1199) in the KO were identified using the same procedure (Appendix A). Lc/nLc- and Gb-series GSLs are potential candidates for H3N2 and H3N2F1 in the KO. In comparison with the MS^2^ reference data of Gb-type STD (H3N2, Forssman), the absence of Gb markers (*m*/*z* 429, 647) in the H3N2 and H3N2F1 KOs led us to anticipate that they would be Lc/nLc-type glycans (Appendix A). In H3N2, digestion with β1-3Galactosidase was observed (change from *m*/*z* 1053 to 891), indicating that it is an Lc-type GSL glycan with β1-3Galactose at the non-reducing terminus (Appendix A). H3N2F1 was cleaved (from *m*/*z* 1199 to 996) by α-*N*-Acetyl-galactosaminidase (αGalNAcase), suggesting the presence of αGalNAc at the non-reducing terminus (Appendix A). For further confirmation, we first cleaved the fucose with α1-3/4Fucosidase to eliminate the peak at *m*/*z* 996 (H3N1F1, originally present), and then treated it with αGalNAcase. This allowed only the structural analysis of the αGalNAcase-digested peak (*m*/*z* 996) (Appendix A). MS^2^ analysis of the GalNAcase digest (H3N1F1, *m*/*z* 996) was performed and compared with the MS^2^ reference data of LNFP I-VI (Appendix A). Since the MS^2^ pattern matched that of LNFP I, the structure of H3N2F1 in the KO was identified as type IA, a blood-type antigen (Appendix A).

Detailed glycan structures were identified in H3N1, H3N1F1, and H3N2F1, and a part of the glycan structure in H3N2, as summarized in Figure 5. These results showed that all GSL glycans that increased in the KO (H3N1, H3N1F1, H3N2, and H3N2F1) had a β1-3Galactosyl structure at the non-reducing terminal. The KO of the galectin-4 in NUGC4 cells resulted in the expression of Lc-type GSL glycans (β1-3Gal).

## 3. Discussion

We investigated the glycan profiles of cell surface proteins and GSLs of the cells with different galectin-4 expression and metastatic potential and found that galectin-4 is involved in the structural changes of GSL glycans in poorly differentiated gastric cancer cells. Detailed structural analysis of the neutral GSL glycans revealed that the addition of β1-3 galactose causes a change in the GSL profile. 

Cancer cells with abnormal proliferative and metastatic potentials often undergo aberrant glycosylation. Therefore, various cancer-associated modifications of glycans are used to diagnose cancer [16,17]. In a previous study, we revealed that galectin-4 participates in the peritoneal dissemination of malignant gastric cancer cells by promoting cell proliferation and interacting with several molecules. More detailed analyses with respect to glycosylation are needed to clarify the mechanisms involved in the regulation of the molecules associated with galectin-4.

In this study, we first analyzed *N*-glycans, because *N*-glycan structures and their biosynthetic glycosyltransferases are often associated with cancer metastasis or suppression [18]. Abundant high-mannose-type glycans (M5, 6, 7, 8, and 9) may be derived from ER and Golgi membrane glycoproteins during *N*-glycan biosynthesis (*m*/*z* 1000–4000) [19]. Wild-type NUGC4 and galectin-4 KO cells mainly have highly fucosylated, highly branched Gal termini and bisected GlcNAc *N*-glycan structures. These glycan structures strongly bind to galectin-4 [20,21]; however, there were no significant differences between the wild type and the KO. 

Because of the multivalency of galectins, members of the galectin family are thought to cross-link surface ligands during the formation of galectin lattices [22]. In NUGC4 cells, the high expression of galectin-4 and galectin-4-binding glycans may result in galectin-4-mediated lattice formation. This may result in abnormal signaling and cell proliferation, which may induce peritoneal dissemination. In the case of a deficiency of an enzyme that forms a multi-antennary type *N*-glycan, reduced levels of molecular interactions between *N*-glycan and galectin on the cell surface are observed [23]. Since *N*-glycan structures of the KO are similar to those of the wild type, this abnormal signaling may decrease because of galectin-4 absence. 

We previously found that galectin-4 binds to GSLs, especially sulfated GSLs [21]. Acidic GSLs were mainly composed of sulfated GSLs because of the absence of sialic acid, as in the case of *N*-glycan (Figure 2). Unexpectedly, we found a significant difference in the neutral GSLs (Figure 3). Since the glycan profiles of the control and rescue cells were similar to those of the wild type, this change was thought to be due to galectin-4 expression and not an artifact from a series of experiments. 

It has been reported that galectin-3 and -4 triggers the GSL-dependent biogenesis of a distinct class of endocytic structures and regulate protein trafficking [24]. Although the mechanism involved in the regulation of the cell surface expression of membrane glycoproteins has not been fully elucidated, the formation of a galectin lattice regulates endocytosis and the residency of surface molecules. Because GSLs are essential ligands for galectin-4, the diversity of GSLs in KO cells may also affect the galectin-4 lattice. 

Detailed structural analysis of the neutral GSL glycans revealed the presence of Lc-type (β1-3galactosyl) glycans in addition to nLc-type (β1-4galactosyl) glycans in the KO (i.e., the β1-3 galactose addition is the cause of the alteration of GSL glycan profiles) (Figure 4). As for H3N1F1 (LNFP II, Le^a^), one of the Lc-type GSL glycans that specifically increased in the KO, it has been reported that it is expressed in the normal human stomach but decreased in gastric adenocarcinoma [25]. Another GSL glycan that showed a specific increase was H3N2F1, a Type IA blood group antigen. The expression of blood group ABO antigens (BGA) has been reported to decrease in gastritis and gastric cancer, suggesting that the loss of BGA expression in gastric epithelial cells is associated with cell differentiation and precancerous changes [26]. 

## 4. Materials and Methods

### 4.1. Cell Lines and Cell Culture

Human gastric cancer NUGC4 and MKN45 cells were obtained from RIKEN BioResource Center (Tsukuba, Japan). Control cells (Con), which were transfected with non-coding crRNA instead of galectin-4 specific crRNA in CRISPR/Cas9-mediated genome-editing and rescue cells (Res), galectin-4 re-expressing cells that were established by transferring the galectin-4-expressing plasmid into a KO clone were established as described previously [15]. Cells were cultured in RPMI 1640 medium (FUJIFILM Wako Pure Chemical, Osaka, Japan) supplemented with 2 mM L-alanyl-L-glutamine solution (FUJIFILM Wako Pure Chemical) and 10% fetal calf serum (FCS) (Funakoshi Co., Ltd., Tokyo, Japan). Cells were washed with phosphate-buffered saline to completely remove FCS and collected by centrifugation at 500× *g* for 5 min to yield cell pellets.

### 4.2. Membrane Protein Extraction from Cells

Membrane proteins were extracted from cells using the CelLytic™ MEM Protein Extraction Kit (Sigma-Aldrich, St. Louis, MO, USA). Cell pellets (1 × 10^7^ cells) were added to 600 μL lysis and separation buffer, pipetted, vortexed, and incubated on ice for 10 min. The cell lysate was then centrifuged (10,000× *g* for 5 min) at 4 °C to remove precipitates. The supernatant of the cell lysate was transferred into a 1.5 mL tube and incubated with a heat block at 30 °C for 5 min. Then, the supernatant of the cell lysate (total protein solubilized solution) was centrifuged (3000× *g*, 3 min) at 25 °C to separate into two layers: upper layer (hydrophilic fraction) and lower layer (hydrophobic fraction, which is the membrane protein fraction). After the upper layer was roughly removed, 400 mL of the wash buffer was added to the lower layer and vortexed. To completely remove the upper layer, the process was repeated after incubation on ice for 10 min, and the lower layer (membrane protein fraction) was collected.

### 4.3. Release of N-Glycans from the Membrane Proteins of Cells

A sample of membrane protein extraction from cells (100–200 μg of protein) was added at 1/10 of the sample volume of 1 M ammonium bicarbonate (Sigma-Aldrich) and 5 μL of 0.2 M dithiothreitol (DTT, Sigma-Aldrich). The sample mixture was reduced by DTT at 60 °C for 30 min followed by alkylation with 5 μL of 0.5 M iodoacetamide (IAA, Sigma-Aldrich) by incubation in the dark at room temperature for 30 min. Then, the sample mixture was incubated at room temperature with 1.25 μL of 0.2 M DTT to remove excess IAA. The sample mixture was then treated with 5 μL of 200 ng/μL trypsin (Sigma-Aldrich) at 37 °C for at least 5 h with shaking (1000 rpm), followed by heat inactivation of the enzyme at 100 °C for 5 min. *N*-Glycans were enzymatically released from trypsin-digested glycopeptides by incubation with 1 μL Rapid PNGase F (New England Biolabs, Ipswich, MA, USA) at 37 °C for at least 16 h with shaking (1000 rpm).

### 4.4. Extraction of GSL from Cells

The GSL extraction procedure was essentially the same as described previously [27]. Briefly, total lipid extracts from cells were sequentially obtained using chloroform/methanol (C/M) solution 2/1, 1/1 (*v*/*v*), and chloroform/methanol/water (C/M/W) solution 1/2/0.8 (*v*/*v*/*v*), by stirring at room temperature. The neutral and acidic fractions were separated by DEAE-Sephadex (A-50) column chromatography (GE Healthcare, Buckinghamshire, UK). The extracted GSLs were separated using HPTLC (Merck, Darmstadt, Germany). Neutral GSLs were developed using chloroform/methanol/water (60/35/8) and visualized using an orcinol reagent. Acidic GSLs were developed using chloroform/methanol/0.22% CaCl_2_ (60/35/8) and visualized using an orcinol reagent. Sulfated GSLs were developed using chloroform/methanol/acetone/acetic acid/water (8/2/4/2/1) and visualized using Azure A reagent. Sulfated GSLs (Cayman Chemical, Ann Arbor, MI, USA, and Matreya LLC, Pleasant Gap, PA, USA), gangliosides (Matreya LLC), and neutral GSLs (Seikagaku Corporation, Tokyo, Japan) were used as standards. For GSL glycan analysis, the supernatants containing crude cellular lipids were completely dried using a centrifugal evaporator and digested with EGCase.

### 4.5. Release of Glycans from the GSL of Cells 

Glycans are enzymatically released from GSL by EGCase [28]. Extracted GSL from 1 × 10^7^ cells was incubated with 5 µL of rEGCase II (Takara Bio, Otsu, Japan) in 45 µL of 20 mM acetate buffer (pH 5.5) at 37 °C for 20 h with shaking (1000 rpm).

### 4.6. Purification and Labeling of N-Glycans and GSL Glycans by Glycoblotting

Purification and labeling of released glycans were performed by glycoblotting using BlotGlyco (Sumitomo Bakelite, Kobe, Japan) as previously reported, with minor modifications [29,30,31,32]. BlotGlyco beads, 50 mL hydrazide-polymer beads slurry (100 mg/mL suspension with water), were placed into the bottom of a reaction tube, and the water was removed by centrifugation (3000× *g*, 1 min). 2% acetic acid (AcOH) in acetonitrile (ACN) of nine times the volume of the sample was applied to the reaction tube with beads followed by the addition of the sample mixtures (20–50 μL) containing released glycans from cells. The reaction tube was incubated at 80 °C for 60 min and dried in a heatblock to capture glycans in the sample mixtures onto beads via stable hydrazone bonds. The reaction tube was washed with 200 μL of 2 M guanidine-HCl, followed by washing with an equal volume of water and 1% triethylamine in methanol (MeOH). Each washing step was performed in duplicate. Unreacted hydrazide functional groups on the beads were capped by incubation with 10% acetic anhydride in methanol at room temperature for 30 min. The solution was removed through centrifugation (3000× *g*, 1 min), and then the beads were washed twice with 200 μL of 10 mM HCl, MeOH, and dimethyl sulfoxide (DMSO), successively. On-bead methyl esterification of the carboxyl groups in sialic acid was conducted via incubation with 100 μL of 500 mM 3-methyl-1-*p*-tolyltriazene (MTT, Tokyo Chemical Industry Co., Ltd., Tokyo, Japan) in DMSO at 60 °C for 60 min. The MTT solution was removed by centrifugation (3000× *g*, 1 min), and the beads were serially washed twice with 200 μL of MeOH and water. For samples without sialic acid, the methyl esterification of sialic acid was excluded. 

For BOA labeling, the glycans blotted on beads were subjected to the transamination reaction with 20 μL of 50 mM *O*-benzylhydroxylamine hydrochloride (Tokyo Chemical Industry Co., Ltd.) and 180 μL of 2% AcOH in ACN for 60 min at 80 °C until completely dry. BOA-labeled glycans were eluted by treatment with 100 μL of water and centrifugation (3000× *g*, 1 min). 

For 2AB labeling, the glycans blotted on beads were subjected to re-release glycans with 20 μL of water and 180 μL of 2% AcOH in ACN for 90 min at 70 °C until completely dry. Re-released glycans were incubated with 50 μL of 2-aminobenzamide (2AB, Tokyo Chemical Industry Co., Ltd.) solution [17.7 mg of 2AB was added to 50 μL AcOH-DMSO-H_2_O (3:1:2), making sure that the reagent was completely dissolved, and then adding 9.1 mg 2-picoline borane (Junsei Chemical Co., Ltd., Tokyo, Japan), making sure that the reagent was completely dissolved] at 50 °C for 60 min. A solution of 2AB-labeled glycans was eluted by centrifugation (3000× *g*, 1 min). Hydrophilic interaction chromatography (HILIC) purification was required for MS analysis of the eluted samples.

### 4.7. HILIC Purification

The sample of eluted 2AB-labeled glycans was added to 1 mL can and mixed by vortexing and pipetting. The mixed sample solution was applied to the clean-up column (amide HILIC column) provided in the BlotGlyco kit, allowed to drop spontaneously for approximately 10 min, and centrifuged (500× *g*, 1 min) to allow complete drainage. The clean-up column with 2AB-labeled glycan sample binding was washed twice with 95% ACN/0.05% FA (600 μL) and centrifuged (3000× *g*, 1 min) to remove the solution. The purified 2AB-labeled glycan sample was eluted twice with 50 μL (total 100 μL) of water by centrifugation (3000× *g*, 1 min).

### 4.8. Fucosidase Digestion

To remove fucose from glycan samples, the purified sample solution of 2AB-GSLs from cells (20 μL→S.V.) and 10 pmol/μL × 2 μL (20 pmol) Control:2AB-labeled LNFP II/III (Seikagaku Corporation) were treated with 20 mU of α1-2 fucosidase (Takara Bio) in 20 mM sodium phosphate buffer (pH 8.6) (total: 10 μL) or with 8 U of α1-3/4 fucosidase (P0769, New England Biolabs) in 20 mM sodium phosphate buffer (pH 6.0) (total: 10 μL) at 37 °C for at least 18 h. If the fucose cleavage of the sample was incomplete, digestion was repeated until complete cleavage was confirmed.

### 4.9. Galactosidase Digestion

To remove galactose from glycan samples, the purified sample solution of 2AB-GSLs from cells (20 μL→S.V.) and Control: de-fucosylated (fucosidase-treated) 10 pmol/μL × 2 μL (20 pmol) 2AB-labeled LNFP II/III (Seikagaku Corporation) were treated with 8 U of β1-4 Galactosidase S (P0745, New England Biolabs) in 20 mM sodium phosphate buffer (pH 6.0) (total: 10–12 μL) or with 10 U of β1-3 Galactosidase (P0726, New England Biolabs) in 20 mM sodium phosphate buffer (pH 6.0) (total: 10–12 μL) at 37 °C for at least 18 h. If galactose cleavage in the sample was incomplete, the digestion was repeated until complete cleavage was confirmed.

### 4.10. Hexosaminidase Digestion

To remove hexosamine from glycan samples, the purified sample solution of 2AB-GSLs from cells (20 μL→S.V.) and 10 pmol/μL × 2 μL (20 pmol) Control: 2AB-H3N2 from forssman antigen (Dia-Iatron Co., Ltd., Tokyo, Japan) were treated with 4 U of β-*N*-Acetyl-glucosaminidase (β1-2,3,4,6) (P0744, New England Biolabs) in 20 mM sodium phosphate buffer (pH 6.0) (total: 10–12 μL), 20 U of α-*N*-Acetyl-galactosaminidase (α1-2,3,4,6) (P0734, New England Biolabs, Japan) in 20 mM sodium phosphate buffer (pH 6.0) (total: 10–12 μL), or 5 U of β-*N*-Acetyl-Hexosaminidase f (β1-3,4,6GlcNAc, β1-4GalNAc) (P0721, New England Biolabs) in 20 mM sodium phosphate buffer (pH 6.0) (total: 10–12 μL) at 37 °C for at least 18 h. If the hexosamine cleavage of the sample was incomplete, digestion was repeated until complete cleavage was confirmed.

### 4.11. MALDI-TOF MS Analysis

For BOA labeled glycans, 0.5–1 μL of sample and 0.5–1 μL a 2,5-dihydroxybenzoic acid (DHB, Shimadzu Biotech, Kyoto, Japan) matrix solution (10 mg/mL DHB in 30% ACN) were mixed on a plate, allowed to dry, and analyzed using MALDI-TOF MS (Autoflex maX, Bruker Daltonics GmbH, Bremen, Germany) [33,34]. BOA labeling in the glycoblotting method has an advantage in that it does not require a post-labeling purification process and allows rapid and simple analysis. However, it is unsuitable for detailed glycan structure analysis using enzyme digestion and/or high performance liquid chromatography (HPLC) because the BOA label is easily dislodged. Therefore, we used 2AB labeling (with reduction) to identify the detailed glycan structures. For 2AB labeled glycans, 0.5–1 μL of sample and 0.5–1 μL DHB matrix solution (10 mg/mL DHB in 30% ACN) were mixed on a plate, allowed to dry, and MS and MS^n^ analyzed using MALDI-QIT-TOF MS (Axima Resonance, Shimadzu Biotech).

For sulfated GSLs, 0.25–0.5 μL of sample and 0.5 μL of α-cyano-4-hydroxycinnamic acid (CHCA, Shimadzu Biotech) matrix solution (Saturated CHCA in 50% ACN) were placed on plates, allowed to dry, and measured by MALDI-TOF MS (Axima TOF^2^, Shimadzu Biotech) and the assignment of the structure was based on previous reports [35]. Sulfated GSLs including the lipid part were detected in negative ion mode MS, because sulfated glycans in acidic GSLs are difficult to detect in the positive ion mode MS. 

*N*-glycans of MKN45 cells were also analyzed by MALDI-TOF MS to confirm the position of peaks of sialylated *N*-glycans (Appendix A).

## 5. Conclusions

In conclusion, our results demonstrate that galectin-4 expression affects the glycan profile of GSLs in malignant cancer cells with a high potential for peritoneal dissemination. These findings suggest that lectins are fundamental regulators of glycosylation in cancer cells. Further detailed analyses are required to determine the precise mechanisms involved in the galectin-4-mediated regulation of associated molecules and glycosylation. Our findings provide novel insights into the regulatory role of galectin-4 that targets several molecules involved in peritoneal dissemination.

## Figures and Tables

**Figure 1 ijms-24-12305-f001:**
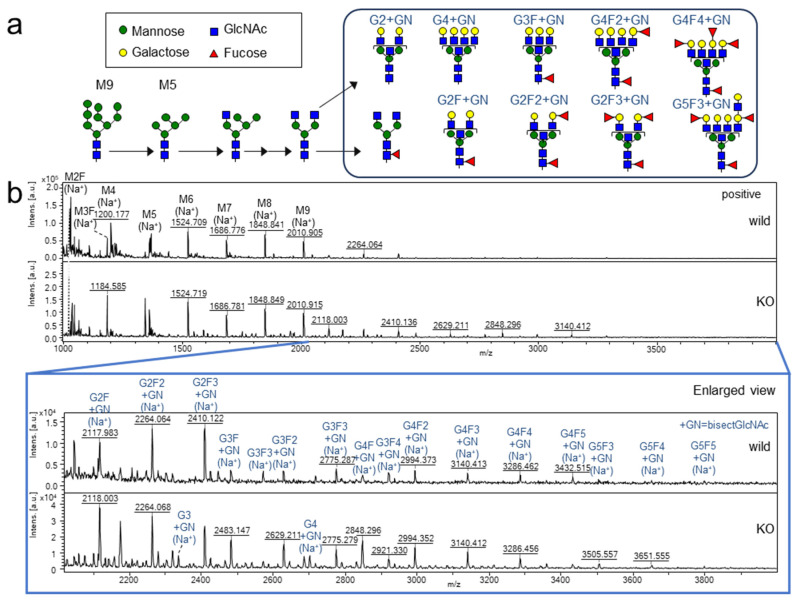
Analysis of the *N*-glycans of the membrane protein from NUGC4 cells by MALDI-TOF MS (BOA-labeling). (**a**) Simple *N*-glycan processing pathway and possible structures of *N*-glycan in NUGC4 cells. (**b**) MALDI-TOF MS spectra (*m*/*z* 1000 to 4000) and enlarged view (*m*/*z* 2020 to 4000) in wild-type NUGC4 (**upper**) and NUGC4 galectin-4 KO (**lower**) were acquired in positive ion mode. The compositions and estimated structures of *N*-glycans are shown in the MS spectra. Some structures were determined by MS/MS analysis (Appendix A).

**Figure 2 ijms-24-12305-f002:**
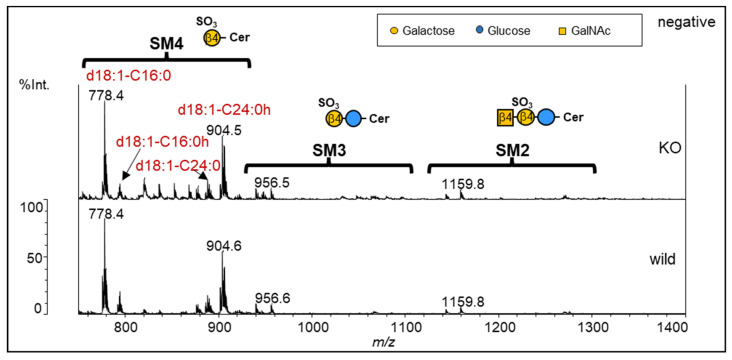
Comparison of MALDI-TOF MS of sulfated GSLs of the wild type (**lower**) and NUGC4 galectin-4 KO (**upper**). MALDI-TOF MS spectra (*m*/*z* 750 to 1400) were acquired in negative ion mode. The estimated structures of sulfated GSLs are shown in the MS spectra.

**Figure 3 ijms-24-12305-f003:**
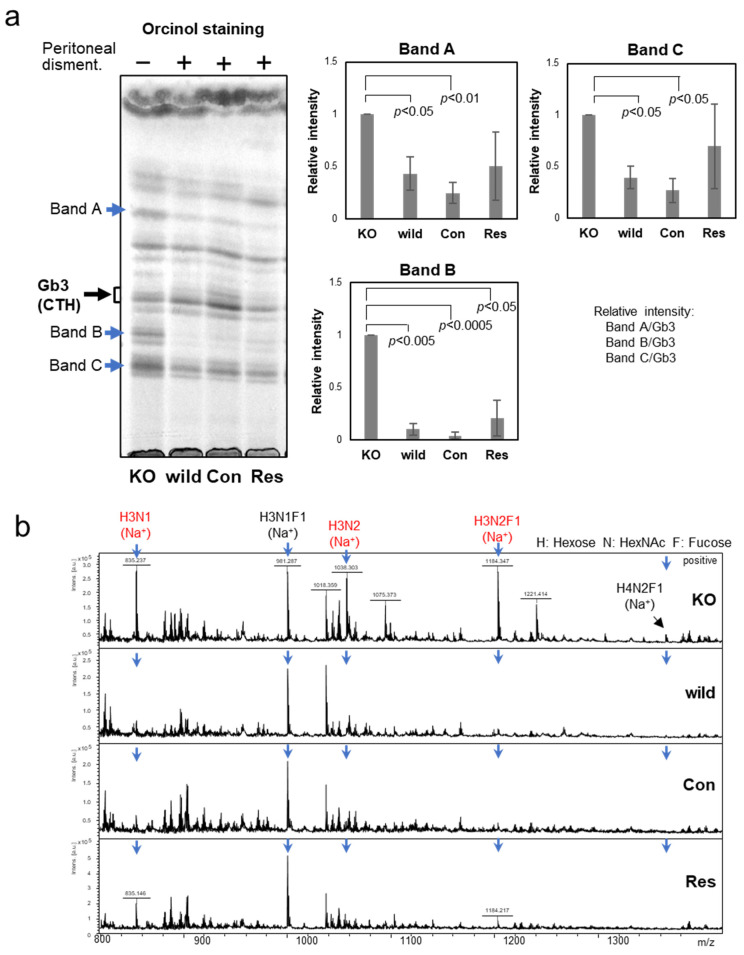
Analysis of neutral GSLs in NUGC4 cells. (**a**) HPTLC analysis of neutral GSLs from NUGC4 cells. HPTLC bands of neutral GSLs were visualized with orcinol staining. Bar graphs represent relative band A-C intensities normalized to that of Gb3 (*n* = 3). (**b**) MALDI-TOF MS analysis of neutral glycans of GSLs from NUGC4 cells (BOA-labeling). MALDI-TOF MS spectra were acquired in positive ion mode. The compositions of neutral glycans of GSLs were shown on the MS spectra.

**Figure 4 ijms-24-12305-f004:**
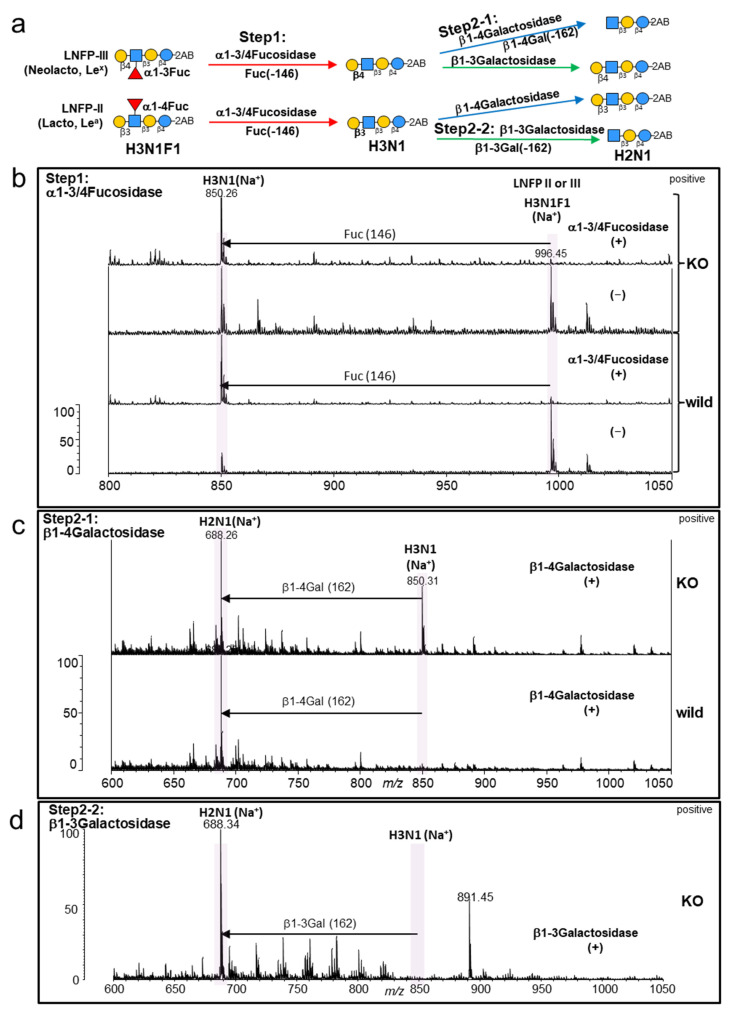
Detailed structure identification of GSL glycans, H3N1F1(*m*/*z* 996) in NUGC4 cells using glycosidase and MS analysis. (**a**) Strategy overview. (**b**) MALDI-TOF MS spectra of GSL glycans with and without α1-3/4Fucosidase in the wild type (**lower two**) and the KO (**upper two**). (**c**) MALDI-TOF MS spectra of GSL glycans with β1-4Galactosidase after α1-3/4Fucosidase in the wild type (**lower**) and the KO (**upper**). (**d**) MALDI-TOF MS spectra of GSL glycans with β1-3Galactosidase after β1-4Galactosidase, α1-3/4Fucosidase in the KO. (**b**–**d**) MALDI-TOF MS spectra of GSL glycans with 2AB labeling were acquired in positive ion mode. The compositions and candidate structures of glycans are shown in the MS spectra.

**Figure 5 ijms-24-12305-f005:**
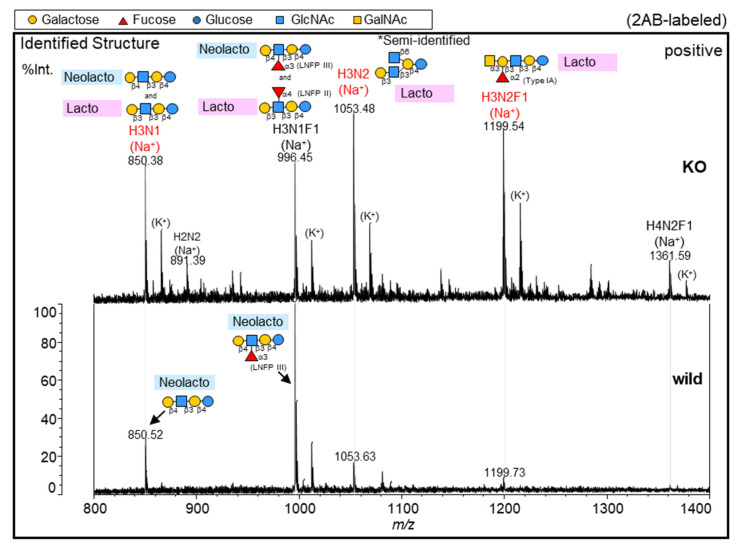
The glycan structures of the neutral glycans of GSLs from NUGC4 cells (wild type and KO). The compositions and identified structures of the glycans of GSLs were shown on the MALDI-TOF MS spectra acquired in positive ion mode. The glycan structures were identified using glycosidase digestion and MS^n^ analysis (Figure 4 and Appendix A).

## Data Availability

The data supporting the findings of this study are available from the corresponding author upon reasonable request.

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
