# Peer review of "Galectin-4 Is Involved in the Structural Changes of Glycosphingolipid Glycans in Poorly Differentiated Gastric Cancer Cells with High Metastatic Potential"

_ijms, 2023, doi:10.3390/ijms241512305_

Round 1

Reviewer 1 Report

The title of the manuscript is remarkable. English language has good quality. Figures need some changes. Results and discussion have written good. There are some modifications that need to be exerted in the citations.

1. Line 29-30 needs proper reference

2. Please rewrite section "Introduction" based on order below:

First: the importance of gastric cancer

Second: the importance of treatment of gastric cancer

Third: the importance of galectin-4 in cancer

Fourth: the importance of your work

3. All multipple and middle sentence references in all over the manuscript should be reformed

4. About the section results:

+ All senyences with reference should be omitted from this section

+ In this section, the authors should only talk about their findings. Any other data that is not a part of results of this study should be excluded from this section

+ Any data about the aim of performing various tests should only mentioned in the part "Material and methods"

+ Any data about the comparison of the results of present research with those in other surveys shohld be mentioned in the section "Discussion"

+ the section "results" should be reformed based on these comments

5. About the section "Discussion"

Please categorize your results based on their importance from the most important one to the least. After that, discuss about each one of them one by one.

6. Please write the section "Conclusion" in a separate part

7. Figure 3-C

All charts should be equal in size

8. Please check and adjust the "Reference list" based on the regulations of reference list of journal. (Titles, doi, the name of journal and ... )

Reviewer 2 Report

Results should be more detail and clear.

Author Response

    Thank you for taking the time to review our manuscript. We have rewritten the results section to be more detailed and easier to understand. We hope that you will consider our revised manuscript suitable for publication. We are open to any further changes that you may deem appropriate.

Reviewer 3 Report

Comments to Authors

The manuscript entitled, “Galectin-4 is involved in the structural changes of glycosphingolipid glycans in poorly differentiated gastric cancer cells with high metastatic potential” by Hachisu et al., showed that galectin-4 regulates glycosylation in gastric cancer cells as glycan profiles of GSLs in malignant cancer cells were altered via galectin-4 expression. A more thorough explanation of their MS methodology and strategy and interpretation, work on figure quality, and more explaining on the processing of N-glycans and GSLs would improve the manuscript for a more general audience. The paper should address the following points:

A major concern is that authors state that sialylated glycans are absent throughout the manuscript. This is quite surprising as earlier papers provide evidence of sialylated N-glycans in NUGC4 cells. An additional method to support the absence of sialylated glycans should be provided or a much thorough explanation for not detecting sialylated glycans. Explain the rationale for using positive ion capture for N-glycans and then negative ion capture for GSLs.

Figure1:

  • Include figure depicting N-glycan processing pathway

Figure 2:

  • Define SM4, SM3, and SM2
  • Include figure depicting GSL processing pathway

Figure 3:

  • Figure 3a:
    • The arrows need to be moved so they do not obstruct the stds
    • Define GM3 and Gb3
    • The region marked for GM3 only partially captures the change that is observed in the KO cell line. What is the upper band?
    • There is a sharp increase in Gb3 binding in the control relative to the other samples. Why is this?
    • There are several bands in the KO that are increased that are ignored, they should be addressed
    • The results should address GM3 and Gb3 analysis, it is not mentioned directly.
  • Figure 3b
    • The doted lines should be removed surrounding the peaks as it makes it difficult to compare intensities. Perhaps use labeled arrows?
    • If there are increases in H3N1, H3N2, H3N2F1, what glycans are being decreased as a result of their synthesis?
  • Figure 4:
    • It would help greatly if this figure could be all on one page.
    • Figure 4a:
      • The strategy should be illustrated more precisely. The figure should show clearly the removal of the respective fucose and galactose residues. Currently, it is not clear whether any fucose or galactose were removed, this should be done clearly to better illustrate this experiment.
    • The MS spectra should be zoomed in on m/z 800-1000 since this is where the changes are occurring. It is difficult to see clearly what is happening
    • Figure 4b
      • In KO comparing the levels of H3N1 with/without digestion. Why is there not a more significant increase in H3N1 following digestion. The amount of H3N1 looks unchanged following fucosidase digestion, while H3N1F is clearly being digested. This change is nicely demonstrated in the WT MS.

Round 2

Reviewer 1 Report

No more comment.

Reviewer 3 Report

Very nice job!

Accept is present form.